# Natural depletion of histone H1 in sex cells causes DNA demethylation, heterochromatin decondensation and transposon activation

Shengbo He, Martin Vickers, Jingyi Zhang, Xiaoqi Feng*

Department of Cell and Developmental Biology, John Innes Centre, Norwich, United Kingdom

**Abstract** Transposable elements (TEs), the movement of which can damage the genome, are epigenetically silenced in eukaryotes. Intriguingly, TEs are activated in the sperm companion cell – vegetative cell (VC) – of the flowering plant *Arabidopsis thaliana*. However, the extent and mechanism of this activation are unknown. Here we show that about 100 heterochromatic TEs are activated in VCs, mostly by DEMETER-catalyzed DNA demethylation. We further demonstrate that DEMETER access to some of these TEs is permitted by the natural depletion of linker histone H1 in VCs. Ectopically expressed H1 suppresses TEs in VCs by reducing DNA demethylation and via a methylation-independent mechanism. We demonstrate that H1 is required for heterochromatin condensation in plant cells and show that H1 overexpression creates heterochromatic foci in the VC progenitor cell. Taken together, our results demonstrate that the natural depletion of H1 during male gametogenesis facilitates DEMETER-directed DNA demethylation, heterochromatin relaxation, and TE activation.

DOI: https://doi.org/10.7554/eLife.42530.001

*For correspondence:
xiaoqi.feng@jic.ac.uk

Competing interests: The authors declare that no competing interests exist.

## Introduction

Large proportions of most eukaryotic genomes are comprised of transposable elements (TEs), mobile genetic fragments that can jump from one location to another. For example, TEs comprise approximately 50% of the human genome (*International Human Genome Sequencing Consortium, 2001*; *Venter et al., 2001*), and more than 85% of the genomes in crops such as wheat and maize (*Schnable et al., 2009*; *Wicker et al., 2018*). Regarded as selfish and parasitic, activities of TEs compromise genome stability, disrupt functional genes, and are often associated with severe diseases including cancers in animals (*Anwar et al., 2017*). To safeguard genome integrity, eukaryotic hosts have evolved efficient epigenetic mechanisms, including DNA methylation, to suppress TEs (*He et al., 2011*; *Law and Jacobsen, 2010*). Curiously, recent studies point to episodes of TE activation that occur in specific cell types and/or particular developmental stages (*Garcia-Perez et al., 2016*; *Martínez and Slotkin, 2012*). These TE activation events provide unique opportunities to understand epigenetic silencing mechanisms, and the co-evolution between TEs and their hosts.

Developmental TE activation has been shown in mammalian embryos, germlines and brain cells. In pre-implantation embryos and the fetal germline, LINE-1 retrotransposons are highly expressed despite relatively low levels of transposition (*Fadloun et al., 2013*; *Kano et al., 2009*; *Percharde et al., 2017*; *Richardson et al., 2017*). Recently, LINE-1 RNA was shown to play a key regulatory role in promoting pre-implantation embryo development in mice (*Percharde et al., 2018*). LINE-1 elements have also been shown to transcribe and mobilize in neuronal precursor cells in mice and human (*Coufal et al., 2009*; *Muotri et al., 2005*). The underlying mechanism of such

**eLife digest** In most organisms, the genetic information contains DNA segments called transposable elements which are able to move around in the genome. When transposable elements insert themselves in a new location, this can lead to negative outcomes such as cell death or cancer.

Animals, plants and other organisms have evolved sophisticated mechanisms to 'lock in' transposable elements and prevent them from jumping from place to place in the genome. For instance, adding small molecules called methyl groups onto these sequences tightly packages the DNA, which wraps itself around proteins known as histones. The resulting structure is known as heterochromatin, and it limits the movement of the transposable elements.

In certain situations, cells may 'reactivate' some of their transposable elements: this is for example the case in plant sperm companion cells, which protect the sperm and deliver them to the egg cell. However, it was not clear how many transposable elements are reactivated in these cells, or how this process works.

Here, He et al. investigate this process in the sperm companion cells of a small weed known as *Arabidopsis thaliana*. The experiments showed that around 100 transposable elements were reactivated, most of them when an enzyme called DEMETER removed the methyl groups found in heterochromatin. However, this enzyme alone was not enough. It could only access the methyl groups if the tightly packed structure of the heterochromatin had relaxed following the removal of a histone protein called H1.

Taken together, these results indicate that histone H1 and DEMETER cooperate to regulate the activity of transposable elements in the genome. In addition, H1 is known to prevent the addition of methyl groups onto DNA; that it also impedes their removal suggests that this protein plays a complex role in controlling the way genetic information is interpreted. The next step would now be to investigate the impact of the reactivation of transposable elements on the next generation of plants and during evolution.

DOI: https://doi.org/10.7554/eLife.42530.002

cell-specific TE activation is still unclear. Hypomethylation at LINE-1 promoters in neurons has been proposed to contribute (*Coufal et al., 2009*), and possibly the availability of transcription factors (*Muotri et al., 2005*; *Richardson et al., 2014*). The frequency of LINE-1 retrotransposition in mammalian brain is still under debate; however, it has been speculated that LINE-1 activities might serve to promote genetic diversity among cells of a highly complex organ like the brain (*Garcia-Perez et al., 2016*; *Richardson et al., 2014*; *Singer et al., 2010*).

One of the best demonstrated cases of developmental TE activation occurs in the male gametophyte of flowering plants, pollen grains. Pollen are products of male gametogenesis, which initiates from haploid meiotic products called microspores. Each microspore undergoes an asymmetric mitosis to generate a bicellular pollen comprised of a large vegetative cell (VC) and a small generative cell engulfed by the VC (*Berger and Twell, 2011*). Subsequently the generative cell divides again mitotically to produce two sperm. Upon pollination, the VC develops into a pollen tube to deliver the sperm to meet the female cells, and subsequently degenerates. In the mature tricellular pollen of *Arabidopsis thaliana*, several TEs were found activated and transposed (*Slotkin et al., 2009*). Enhancer/gene trap insertions into TEs showed specific reporter activity in the VC, and TE transpositions detected in pollen were absent in progeny (*Slotkin et al., 2009*). These results demonstrated TE activation in pollen is confined to the VC. TE expression in the short-lived VC has been proposed to promote the production of small RNAs, which may be transported into sperm to reinforce the silencing of cognate TEs (*Calarco et al., 2012*; *Ibarra et al., 2012*; *Martínez et al., 2016*; *Slotkin et al., 2009*). However, TE transcription in the VC has not been comprehensively investigated, hindering our understanding of this phenomenon.

The mechanisms underlying TE reactivation in the VC are also unknown. One proposed mechanism is the absence of the Snf2 family nucleosome remodeler DDM1 (*Slotkin et al., 2009*). DDM1 functions to overcome the impediment of nucleosomes and linker histone H1 to DNA methyltransferases (*Lyons and Zilberman, 2017*; *Zemach et al., 2013*). Loss of DDM1 leads to DNA hypomethylation and massive TE derepression in somatic tissues (*Jeddeloh et al., 1999*; *Lippman et al.,*

*2004*; *Tsukahara et al., 2009*; *Zemach et al., 2013*). However, global DNA methylation in the VC is comparable to that of microspores and substantially higher than in somatic tissues (*Calarco et al., 2012*; *Hsieh et al., 2016*; *Ibarra et al., 2012*). This suggests that DDM1 is present during the first pollen mitosis that produces the VC, so its later absence is unlikely to cause TE activation.

A plausible mechanism underlying TE activation in the VC is active DNA demethylation. DNA methylation in plants occurs on cytosines in three sequence contexts: CG, CHG and CHH (H = A, C or T). Approximately ten thousand loci – predominantly TEs – are hypomethylated in the VC, primarily in the CG context and to a lesser extent in the CHG/H contexts (*Calarco et al., 2012*; *Ibarra et al., 2012*). Hypomethylation in the VC is caused by a DNA glycosylase called DEMETER (DME) (*Ibarra et al., 2012*). DME demethylates DNA via direct excision of methylated cytosine, and its expression is confined to the VC and its female counterpart, the central cell, during sexual reproduction (*Choi et al., 2002*; *Schoft et al., 2011*). DME demethylation may therefore cause TE transcription in the VC; however, this hypothesis has not been tested.

Another plausible mechanism for epigenetic TE activation is chromatin decondensation (*Feng et al., 2013*). Drastic reprogramming of histone variants and histone modifications occurs during both male and female gametogenesis, rendering the gametes and companion cells with radically different chromatin states (*Baroux et al., 2011*; *Borg and Berger, 2015*). For example, centromeric repeats, which are condensed in sperm and other cell types, are decondensed in the VC, accompanied by the depletion of centromeric histone H3 (*Ingouff et al., 2010*; *Mérai et al., 2014*; *Schoft et al., 2009*). Chromocenters, which are comprised of condensed pericentromeric heterochromatin and rDNA repeats (*Chandrasekhara et al., 2016*; *Fransz et al., 2002*; *Tessadori et al., 2004*), are observed in sperm nuclei but absent in the VC nucleus, suggesting that pericentromeric heterochromatin is decondensed in the VC (*Baroux et al., 2011*; *Ingouff et al., 2010*; *Schoft et al., 2009*). Heterochromatin decondensation in the VC is proposed to promote rDNA transcription that empowers pollen tube growth (*Mérai et al., 2014*). However, the cause of VC heterochromatin decondensation remains unclear.

Our previous work showed that histone H1, which binds to the nucleosome surface and the linker DNA between two adjacent nucleosomes (*Fyodorov et al., 2018*), is depleted in *Arabidopsis* VC nuclei (*Hsieh et al., 2016*). H1 depletion in the VC has also been observed in a distantly related lily species (*Tanaka et al., 1998*), suggesting a conserved phenomenon in flowering plants. In *Drosophila* and mouse embryonic stem cells, H1 has been shown to contribute to heterochromatin condensation (*Cao et al., 2013*; *Lu et al., 2009*). H1 is also more abundant in heterochromatin than euchromatin in *Arabidopsis* (*Ascenzi and Gantt, 1999*; *Rutowicz et al., 2015*). However, it is unknown whether H1 participates in heterochromatin condensation in plant cells, and specifically whether the lack of H1 contributes to heterochromatin decondensation in the VC.

Whether and how the depletion of H1 in the VC contributes to TE derepression is also unclear. A recent study pointed to an intriguing link between H1 and DME. In the central cell, the histone chaperone FACT (facilitates chromatin transactions) is required for DME-directed DNA demethylation in heterochromatic TEs, and this requirement is dependent on H1 (*Frost et al., 2018*). However, DME activity in the VC is independent of FACT (*Frost et al., 2018*). One attractive hypothesis is that the lack of H1 in the VC causes heterochromatin decondensation and thereby contributes to the independence of DME from FACT. H1 depletion may therefore participate in VC TE activation by promoting DME-directed demethylation. Additionally, H1 depletion may activate TE transcription independently of DNA methylation, as shown in *Drosophila* where DNA methylation is absent (*Iwasaki et al., 2016*; *Lu et al., 2013*; *Vujatovic et al., 2012*; *Zemach et al., 2010*; *Zhang et al., 2015*).

In this study, we identify heterochromatic TEs that are epigenetically activated in *Arabidopsis* VCs. We demonstrate that these TEs are typically subject to DME-directed demethylation at the transcriptional start site (TSS), which is at least partially permitted by the depletion of H1. However, we find that loss of H1 activates some TEs without altering DNA methylation. We also show that developmental depletion of H1 decondenses heterochromatin in late microspores and is important for pollen fertility. Our results demonstrate that H1 condenses heterochromatin in plants and maintains genome stability by silencing TEs via methylation-dependent and -independent mechanisms.

## Results

### Heterochromatic transposons are preferentially expressed in the vegetative cell

To measure the extent of TE activation in the VC, we performed RNA-seq using mature pollen grains, followed by the annotation of gene and TE transcripts using Mikado and the TAIR10 annotation (*Venturini et al., 2018*). We identified 114 TEs that are transcribed at significantly higher levels in pollen than rosette leaves (fold change >2; p<0.05, likelihood ratio test), and hence likely to be specifically activated in the VC (*Figure 1—source data 1*) (*Slotkin et al., 2009*).

The VC-activated TEs are primarily located in pericentromeric regions and exhibit features of heterochromatic TEs, such as being long and GC rich (*Frost et al., 2018*) (*Figure 1A,B*, *Figure 1—figure supplement 1A*). As is typical of heterochromatic TEs (*Zemach et al., 2013*), VC-activated TEs are significantly enriched in dimethylation of histone H3 on lysine 9 (H3K9me2) in somatic tissues, and are significantly depleted of euchromatin-associated modifications (*Figure 1B*, *Figure 1—figure supplement 1B*). VC-activated TEs encompass diverse TE families, among which MuDR DNA transposons and Gypsy LTR-retrotransposons are significantly overrepresented (p<$10^{-9}$ and 0.01, respectively, Fisher's exact test; *Figure 1C*).

### Transposon derepression in the VC is caused by DME-directed DNA demethylation

To assess whether TE activation in the VC is caused by DME-mediated DNA demethylation, we examined DNA methylation in VC and sperm at the 114 activated TEs. We found that these TEs have substantially lower CG methylation in the VC than in sperm at and near the TSS (*Figure 1D,E*), indicative of DME activity. Because TEs tend to be flanked by repeats (*Joly-Lopez and Bureau, 2018*), the transcriptional termination site (TTS) regions of activated TEs also tend to be hypomethylated in the VC (*Figure 1D,E*, *Figure 1—figure supplement 1C*). Examination of DNA methylation in VCs from *dme/+* heterozygous plants (*dme* homozygous mutants are embryonic lethal), which produce a 50:50 ratio of *dme* mutant and WT pollen, revealed an intermediate level of methylation at TSS and TTS of VC-activated TEs (*Figure 1D,E*). CHG and CHH methylation is also substantially increased at the TSS (and TTS) of VC-activated TEs in *dme/+* VC (*Figure 1—figure supplement 1C*), consistent with the knowledge that DME demethylates all sequence contexts (*Gehring et al., 2006*; *Ibarra et al., 2012*).

Consistent with the above results, 71 of the 114 (62%) VC-activated TEs overlap VC DME targets at their TSSs (*Figure 1F*, *Figure 1—source datas 1* and *2*). 92 out of the 114 TEs (81%) have VC DME targets within 500 bp of the TSS (*Figure 1F*, *Figure 1—figure supplement 1D*, *Figure 1—source data 1*). As DNA methylation at/near the TSS has been well-demonstrated to suppress the transcription of genes and TEs in plants and animals (*Barau et al., 2016*; *Eichten et al., 2012*; *Hollister and Gaut, 2009*; *Manakov et al., 2015*; *Meng et al., 2016*), our results indicate that DME-directed demethylation is a major mechanism of TE activation in the VC. Consistently, RNA-seq of pollen from *dme/+* heterozygous plants showed significantly reduced expression at the 114 VC-activated TEs (*Figure 1G*, *Figure 1—source data 1*). As *dme/+* heterozygous plants produce half *dme* mutant and half WT pollen, we expect the transcription of VC-activated TEs to be reduced to roughly half in *dme/+* pollen. The 114 VC-activated TEs are transcribed at levels close to this expectation (*Figure 1G*), with the median ratio of their transcription in *dme/+* versus WT pollen being 0.43 (*Figure 1—source data 1*).

### Vegetative-cell-expressed H1 impedes DME from accessing heterochromatic transposons

We next tested our hypothesis that the lack of histone H1 in the VC (*Hsieh et al., 2016*) allows heterochromatin to be accessible by DME. We first examined the developmental timing of H1 depletion during microspore and pollen development using GFP translational fusion lines (*Hsieh et al., 2016*; *She et al., 2013*). There are three H1 homologs in *Arabidopsis*, with H1.1 and H1.2 encoding the canonical H1 proteins, and H1.3 expressed at a much lower level and induced by stress (*Rutowicz et al., 2015*). H1.1- and H1.2- GFP reporters exhibit the same expression pattern: present in early microspore nucleus but absent in the late microspore stage, and remaining absent in the VC

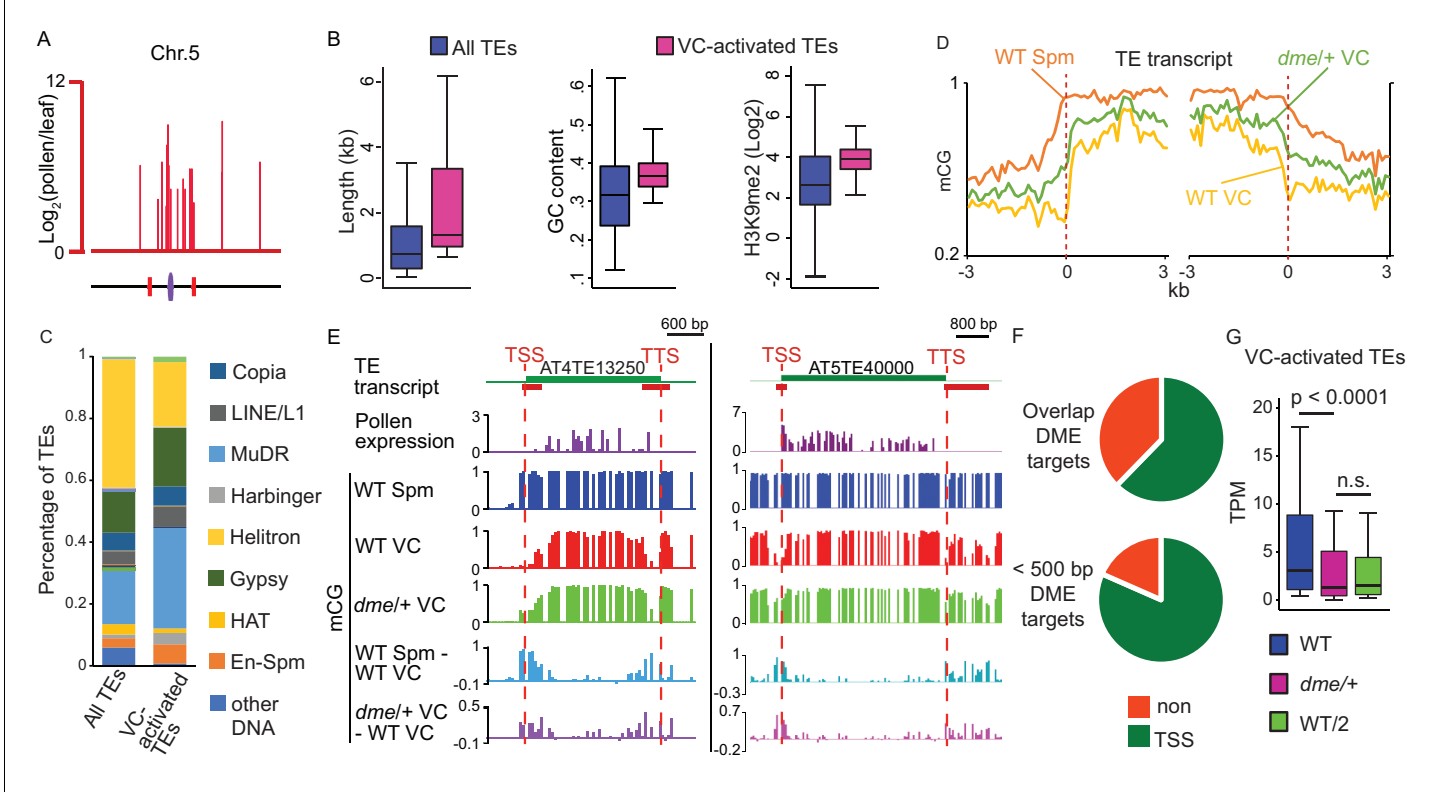

**Figure 1.** VC-activated TEs are heterochromatic and demethylated by DME. (**A**) Expression and locations of VC-activated TEs along Chromosome 5. The purple ellipse and red bars indicate the centromere and borders of pericentromeric regions, respectively. (**B**) Box plots showing the length, GC content, and H3K9me2 level of TEs. Each box encloses the middle 50% of the distribution, with the horizontal line marking the median and vertical lines marking the minimum and maximum values that fall within 1.5 times the height of the box. Difference between the two datasets compared for each feature is significant (Kolmogorov-Smirnov test p<0.001). (**C**) Percentages of TEs classified by superfamily. (**D**) VC-activated TEs were aligned at the TSS and TTS (dashed lines), respectively, and average CG methylation levels for each 100 bp interval were plotted (referred to as ends analysis). (**E**) Snapshots demonstrating the expression (Log$_2$RPKM), absolute and differential CG methylation at two example VC-activated TEs. Red lines under TE annotations indicate VC DME targets. Spm, sperm. (**F**) Pie charts illustrating percentages of VC-activated TEs with TSS overlapping (top) or within 500 bp (bottom) of VC DME targets. (**G**) Box plot showing the expression level of VC-activated TEs in pollen from WT and *dme/+* heterozygous mutant. WT/2 shows half of the WT expression level. Wilcoxon matched-pairs signed-rank test is used. n.s., no significance.

DOI: https://doi.org/10.7554/eLife.42530.003

The following source data and figure supplement are available for figure 1:

**Source data 1.** List of VC-activated TEs.
DOI: https://doi.org/10.7554/eLife.42530.005
**Source data 2.** List of VC DME targets.
DOI: https://doi.org/10.7554/eLife.42530.006
**Figure supplement 1.** VC-activated TEs are heterochromatic and demethylated by DME.
DOI: https://doi.org/10.7554/eLife.42530.004

nucleus while present in the generative cell and subsequent sperm nuclei (*Figure 2A*). H1.3 is not detectable in either microspore or pollen (*Figure 2A*). These results are consistent with our previous observations, confirming that H1 is absent in the VC (*Hsieh et al., 2016*), and demonstrating that H1 depletion begins at the late microspore stage.

To understand how H1 affects DME activity, we ectopically expressed H1 in the VC. To ensure H1 incorporation into VC chromatin, we used the *pLAT52* promoter, which is expressed from the late microspore stage immediately prior to Pollen Mitosis 1, and is progressively upregulated in VC during later stages of pollen development (*Eady et al., 1994*; *Grant-Downton et al., 2013*). Using *pLAT52* to drive the expression of H1.1 tagged with mRFP (simplified as *pVC::H1*), we observed continuous H1-mRFP signal in the VC at the bicellular and tricellular pollen stages, while the signal was undetectable in the generative cell and sperm (*Figure 2B*). H1-mRFP signal was also undetectable in

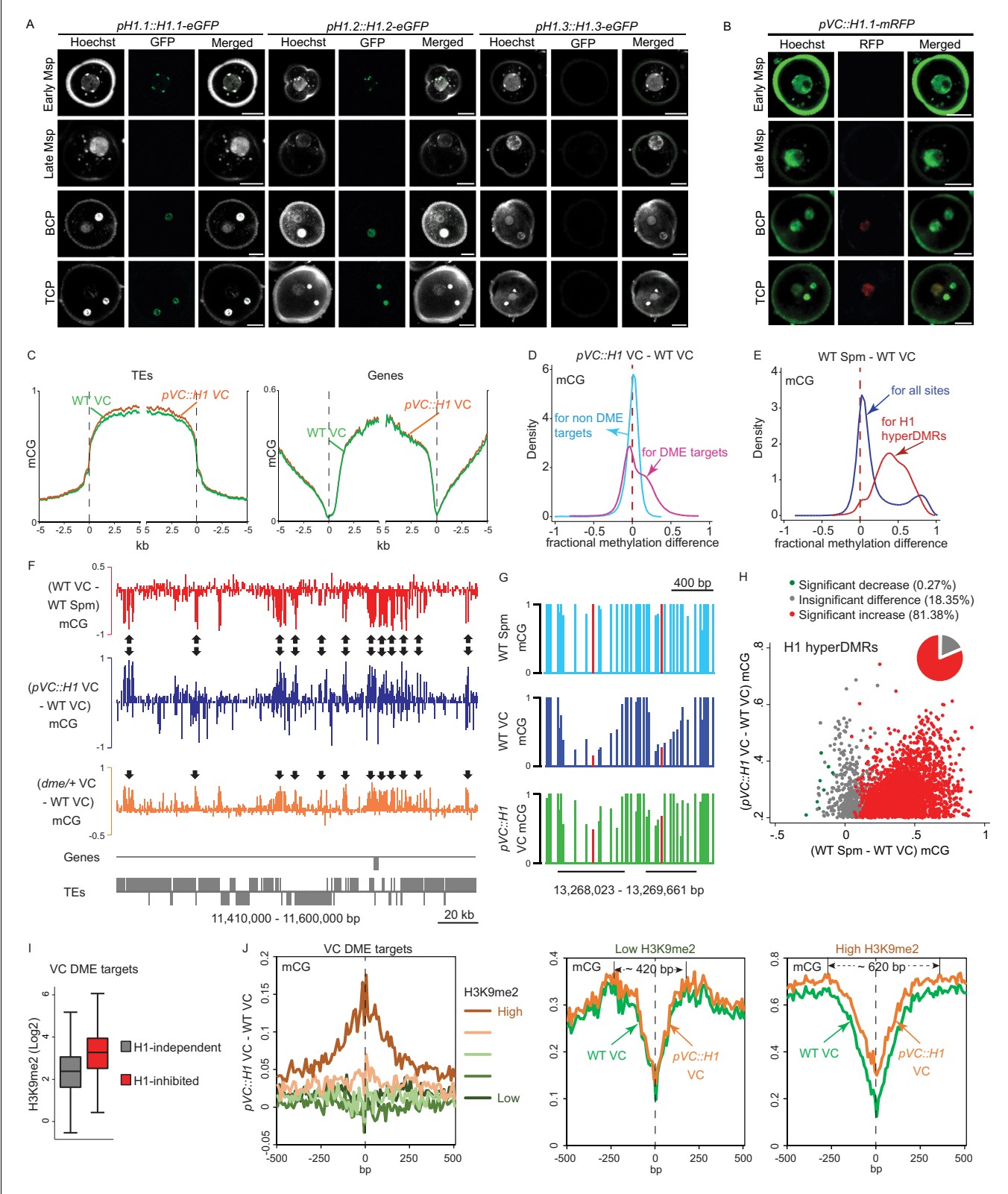

**Figure 2.** Ectopic H1 expression in the vegetative cell impedes DME at the most heterochromatic loci. (A–B) Confocal images showing H1 localization under native promoter (A) and VC-specific promoter (pVC, B) during male gametogenesis. Msp, microspore; BCP, bicellular pollen; TCP, tricellular pollen. Bars, 5 μm. All pVC::H1.1-mRFP (short as pVC::H1) refers to line #2. (C) Ends analysis of all TEs or genes in VCs from pVC::H1 (line #2) and WT. (D–E) Kernel density plots illustrating frequency distribution of methylation differences in 50 bp windows between VCs from pVC::H1 and WT (D), and

*Figure 2 continued on next page*

*Figure 2 continued*

between WT sperm (Spm) and VC (E). (F) Snapshots showing CG methylation difference between the indicated cell types. Arrows point to DME targets that are hypermethylated by *pVC::H1*. (G) Snapshots demonstrating CG methylation in sperm and VCs at single-nucleotide resolution, with the cytosine most hypomethylated by DME marked in red. VC DME targets are underlined in black. (H) Scatter plot illustrating CG methylation differences between the indicated cell types at H1 hyperDMRs. 82.25% of H1 hyperDMRs show significant increase in sperm in comparison to VCs. (I) Box plot illustrating H3K9me2 level at VC DME targets that are significantly hypermethylated in *pVC::H1* (H1-inhibited) or not (H1-independent), respectively. Difference between the two groups is significant (Kolmogorov-Smirnov test p<0.001). (J) VC DME targets were grouped according to H3K9me2 levels, aligned at the most demethylated cytosine (dashed lines), and plotted for average CG methylation difference as indicated in each 10 bp interval (left). Similarly, CG methylation in *pVC::H1* and WT VCs was plotted for the group with the lowest and highest H3K9me2, respectively. Spm, sperm.

DOI: https://doi.org/10.7554/eLife.42530.007

The following source data and figure supplements are available for figure 2:

**Source data 1.** List of H1 hyperDMRs.

DOI: https://doi.org/10.7554/eLife.42530.010

**Figure supplement 1.** H1 ectopic expression in the vegetative cell causes pollen defect and reduced fertility.

DOI: https://doi.org/10.7554/eLife.42530.008

**Figure supplement 2.** H1 ectopic expression in the vegetative cell causes DNA hypermethylation at DME targets.

DOI: https://doi.org/10.7554/eLife.42530.009

late microspores (*Figure 2B*), probably due to the low activity of *pLAT52* at this stage (*Eady et al., 1994*). Notably, we found H1 expression in VC leads to shortened siliques, a substantial proportion of malformed pollen, and reduced pollen germination rate (*Figure 2—figure supplement 1A–D*), suggesting the depletion of H1 in the VC is important for pollen fertility.

To evaluate the effect of VC-expressed H1 on DNA methylation, we obtained genome-wide methylation profiles for VC nuclei from a strong *pVC::H1* line (#2; *Figure 2B*) and WT via fluorescence-activated cell sorting (FACS) followed by bisulfite sequencing (*Supplementary file 1*). CG methylation in the VC of *pVC::H1* plants is largely similar to that of WT, except for a slight increase in TE methylation (*Figure 2C*, *Figure 2—figure supplement 2A*). Consistently, the frequency distribution of CG methylation differences between VCs of *pVC::H1* and WT at loci that are not DME targets peaks near zero, showing almost no global difference (*Figure 2D*). However, a substantial proportion of loci that are targeted by DME show hypermethylation in *pVC::H1* VC (*Figure 2D*, *Figure 2—figure supplement 2B*). DME targets also show preferential hypermethylation in CHG and CHH contexts in the VC of *pVC::H1* (*Figure 2—figure supplement 2C–D*). These results indicate that H1 expression in the VC specifically impedes DME activity.

Across the genome, we found 2964 differentially methylated regions (DMRs) that are significantly CG hypermethylated in the VC of *pVC::H1* plants (referred to as H1 hyperDMRs hereafter; ranging from 101 to 2155 nt in length, 280 nt on average; *Figure 2—source data 1*). Most of the H1 hyperDMRs (1618, 55%) overlap DME targets in the VC (*Figure 2—source data 1*), and H1 hyperDMRs exhibit strong hypomethylation in WT VCs, with 81.4% (2412 sites) having significantly more CG methylation in sperm than VC (p<0.001, Fisher's exact test), indicating that most H1 hyperDMRs are DME targets (*Figure 2E–H*).

Our results demonstrate that H1 hyperDMRs are primarily caused by the inhibition of DME. However, only 3066 out of 11896 (26%) VC DME targets have significantly more CG methylation in the VC of *pVC::H1* than WT (p<0.001, Fisher's exact test; *Figure 1—source data 2*), indicating that VC-expressed H1 impedes DME at a minority of its genomic targets. These H1-impeded DME targets are heterochromatic and significantly enriched in H3K9me2 compared with H1-independent DME targets (*Figure 2I*). To further examine the link with heterochromatin, we aligned all VC DME target loci at the most hypomethylated cytosine, and separated them into five groups by H3K9me2 levels (*Figure 2J*). *pVC::H1*-induced hypermethylation peaks where DME-mediated hypomethylation peaks, but is apparent only in the most heterochromatic group (highest H3K9me2) of DME target loci (*Figure 2J*). Taken together, our results demonstrate that developmental removal of H1 from the VC allows DME to access heterochromatin.

## H1 represses transposons via methylation-dependent and independent mechanisms

Given the importance of H1 removal for DME-directed DNA demethylation and the well-demonstrated role of DME demethylation in regulating gene expression (*Choi et al., 2002*; *Ibarra et al., 2012*; *Schoft et al., 2011*), we investigated the contribution of H1 to gene regulation in pollen. RNA-seq was performed using pollen from the *pVC::H1* line (#2), which showed strong H1 expression in VC (*Figures 2B* and *3A*). Only a small fraction of pollen-expressed genes (3%; 89 out of 2845) is differentially expressed (fold change >2; p<0.05, likelihood ratio test) between *pVC::H1* and WT (*Figure 3B*, *Figure 3—source data 1*). Among these 89 genes, 58 (65%) are suppressed by H1 expression in the VC, and 31 (35%) are activated (*Figure 3B*, *Figure 3—source data 1*). 85 out of these 89 genes (96%) do not overlap DME targets within 500 bp of the TSS, hence the effect of H1 on their expression is probably not mediated by DME. Among the four genes that overlap DME targets within 500 bp of the TSS, two genes gain a small amount of methylation at the overlapping DME targets and are suppressed in *pVC::H1* pollen (*Figure 3—figure supplement 1*, *Figure 3— source data 1*), hence are possibly suppressed by H1 via the inhibition of DME.

Because DME regulates the expression of imprinted genes in the endosperm (*Choi et al., 2002*; *Hsieh et al., 2011*; *Ibarra et al., 2012*), we specifically examined the imprinted genes to further investigate the role of VC H1 removal in regulating pollen gene transcription. Out of the 640 known imprinted genes (*Gehring et al., 2011*; *Hsieh et al., 2011*; *Pignatta et al., 2014*; *Wolff et al., 2011*), 85 overlap endosperm DME targets within 500 bp of their TSS (*Figure 3—source data 2*). Because DME targets different loci in the VC and CC (*Ibarra et al., 2012*), we subsequently examined if these 85 putative DME-regulated imprinted genes are also subject to DME demethylation in the VC. 63 of these genes also overlap VC DME targets within 500 bp of their TSS, none of which is differentially expressed in *pVC::H1* pollen compared with WT pollen (*Figure 3—source data 2*), showing that transcription of these genes in pollen is unlikely regulated by H1. This is not surprising, because only 8 out of these 63 genes are expressed in WT pollen (*Figure 3—source data 2*), and H1 preferentially regulates heterochromatic DME targets (*Figures 2I,J* and *3G*), whereas DME targets involved in gene regulation are typically euchromatic sites next to genes (*Figure 3C*). Consistently, DME targets likely regulating imprinted genes are associated with significantly less H1 in somatic tissues than the VC DME targets that are dependent on H1 (*Figure 3C*).

In contrast to the small effect of H1 on gene transcription, a substantial proportion of VC-activated TEs (41%; 47 out of 114) show significant differential expression (fold change >2; p<0.05, likelihood ratio test) due to H1 expression in VC (*Figure 3D*). Among these differentially expressed TEs, the overwhelming majority (46; 98%) are repressed (*Figure 3D,E*, *Figure 1—source data 1*). These data indicate that ectopic expression of H1 in the VC preferentially represses TEs. Quantitative RT-PCR validated our RNA-seq results and confirmed the strong suppression of TEs in *pVC::H1* (*Figure 3F*). Taking advantage of a *pVC::H1* line #7 with weaker H1 expression in pollen (*Figure 3A*), we found H1 represses TE expression in a dosage-dependent manner: TEs are suppressed to a lesser extent in line #7 compared to the strong line #2 (*Figure 3F*). H1-repressed TEs in the VC are predominantly localized to pericentromeric regions and are overrepresented for LTR retrotransposons, including Gypsy and Copia elements (*Figure 3G–I*). Compared to other VC-activated TEs, the H1-repressed TEs are significantly longer and enriched for H3K9me2 and H1 in somatic tissues (*Figure 3H*), consistent with the observation that H1 precludes DME access to heterochromatin.

In support of the hypothesis that H1 represses VC TE expression by blocking DME, 20 of 46 H1-repressed TEs show significant increase of DNA methylation in at least one sequence context within 500 bp of the TSS in *pVC::H1* (p<0.001, Fisher's exact test; *Figure 4A,B*). Four more TEs overlap a DME target, which is hypermethylated in *pVC::H1*, within 500 bp of the TSS, and hence may also be suppressed by DME inhibition. However, 21 out of the rest of 22 TEs do not overlap any H1 hyperDMRs within 500 bp of the TSS (*Figure 4A*, marked by asterisks in the lower panel), indicating that their suppression by H1 is not mediated by DNA methylation. Of these 21 TEs, 15 TEs overlap DME targets within 500 bp of TSS. DME maintains access to these TEs in the presence of H1, suggesting their VC demethylation does not rely on the depletion of H1 and their repression in *pVC::H1* is DME-independent as exemplified by AT3TE60310 (*Figure 4C*). Our results demonstrate that H1 overexpression in the VC represses heterochromatic TEs via both DNA methylation-dependent and independent mechanisms.

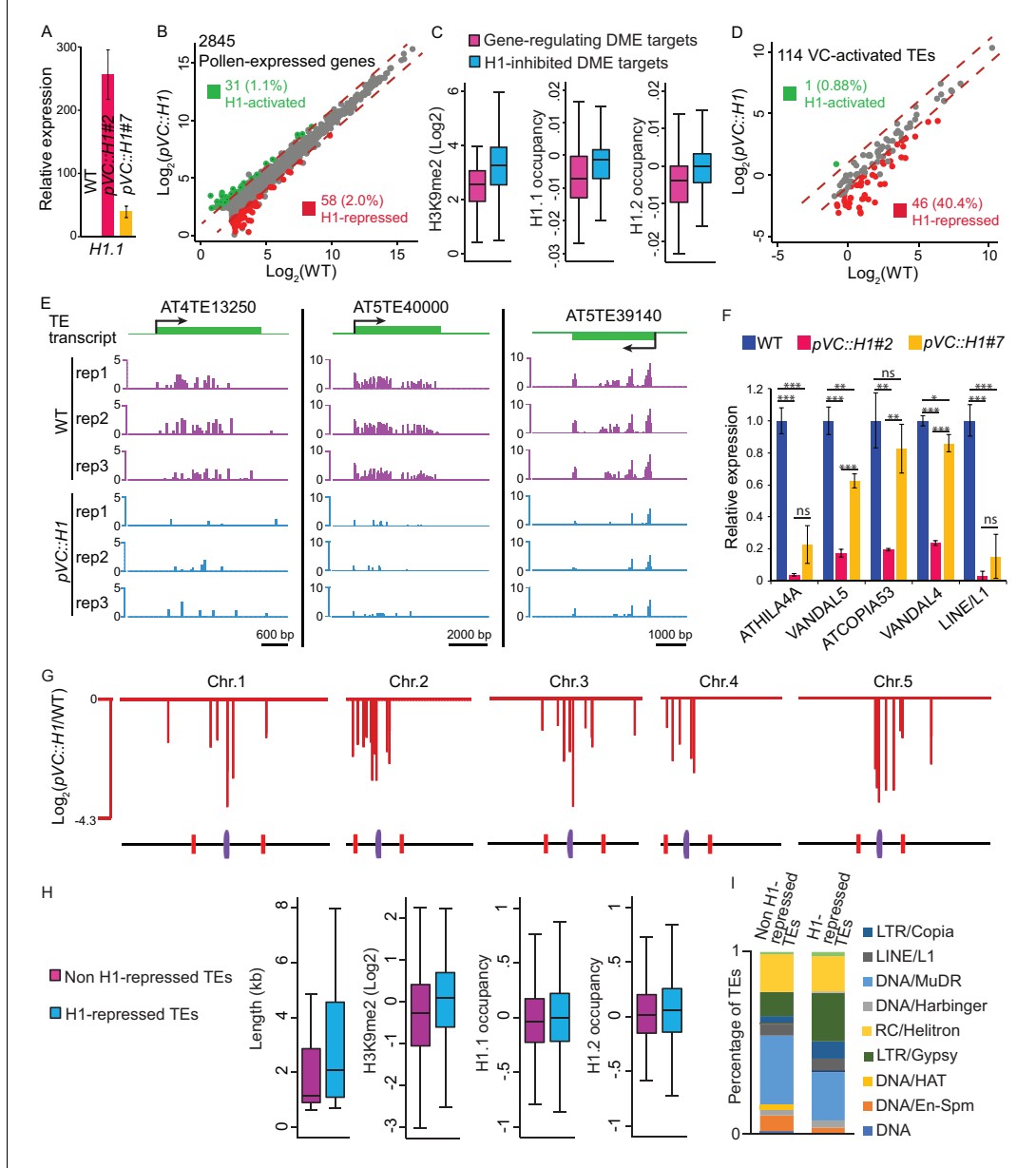

**Figure 3.** Vegetative-cell-expressed H1 represses heterochromatic TEs in a dosage-dependent manner. *pVC::H1* refers to line #2 except as specified in A) and F). (A,F) quantitative RT-PCR demonstrating *H1.1* (A) or TE (F) expression in pollen from WT and two independent *pVC::H1* transgenic lines. Relative expression is calculated by normalizing to WT (WT = 1). Student's *t* test *p<0.05, **p<0.01, ***p<0.001; ns, not significant; n = 3; mean ± SD are shown. (B,D) Scatter plot illustrating the expression (Log2TPM) of TEs or genes in WT and *pVC::H1* pollen. Red and green dots indicate significant down- and up-regulation in *pVC::H1* compared to WT (|fold change| > 2, marked by dashed lines; likelihood ratio test p<0.05), respectively. (C) Box plots illustrating H3K9me2 enrichment and H1 occupancy at endosperm DME targets which are within 500 bp of the TSS of imprinted genes, or VC DME targets inhibited by *pVC::H1* (refer to **Figure 2I**). Difference between the two datasets compared for each feature is significant (Kolmogorov-Smirnov test p<0.001). (E) Snapshots showing the expression (Log2RPKM) of 3 example H1-repressed TEs in WT and *pVC::H1* pollen. Rep, biological replicate. (G) Chromosomal view of H1-repressed TEs, similar to **Figure 1A**. (H) Box plots illustrating the length, H3K9me2 enrichment, and H1 occupancy at two groups of VC-activated TEs. Difference between the two datasets compared for each feature is significant (Kolmogorov-Smirnov test p<0.05 for length, and <0.001 for others). (I) Percentages of TEs classified by superfamily.

DOI: https://doi.org/10.7554/eLife.42530.011

The following source data and figure supplement are available for figure 3:

**Source data 1.** List of pollen-expressed genes and their expression in WT and *pVC::H1* pollen.
DOI: https://doi.org/10.7554/eLife.42530.013
**Source data 2.** List of imprinted genes and their expression in WT and *pVC::H1* pollen.

*Figure 3 continued on next page*

*Figure 3 continued*

DOI: https://doi.org/10.7554/eLife.42530.014

**Figure supplement 1.** Two genes suppressed by VC-expressed H1 that gain methylation at nearby DME targets.

DOI: https://doi.org/10.7554/eLife.42530.012

## Depletion of H1 decondenses heterochromatin during male gametogenesis

H1 depletion and TE activation in the VC are accompanied by loss of cytologically detectable heterochromatin (*Baroux et al., 2011*; *Ingouff et al., 2010*; *Schoft et al., 2009*). We therefore tested whether H1 contributes to heterochromatin condensation in plant cells. Immunostaining of leaf nuclei showed that H1 co-localizes with H3K9me2 in highly-compacted heterochromatic foci, known as chromocenters (*Figure 5A*). Furthermore, we found that chromocenters become dispersed in the nuclei of *h1* mutant rosette leaves (*Figure 5B*). These observations demonstrate that H1 is required for heterochromatin condensation in plants.

We then examined whether ectopic H1 expression can condense the heterochromatin in VC nuclei. Consistent with previous observations (*Baroux et al., 2011*; *Ingouff et al., 2010*; *Schoft et al., 2009*), no condensed chromocenters were detected in WT VC (*Figure 5C*). *pVC::H1* VC also showed no obvious chromocenters (n > 500; *Figure 2B*). This suggests either that H1 expression is not strong enough in *pVC::H1*, or other factors are involved in heterochromatin decondensation in the VC.

Heterochromatin decondensation during male gametogenesis seems to be gradual: chromocenters are observed at early microspore stage, but become dispersed in late microspore stage, when H1 is depleted (*Figures 2A* and *5C*). We observed strong and weak chromocenters, respectively, in 27% and 59% of late microspore nuclei, whereas no chromocenters were observed in the VC at either bicellular or tricellular pollen stage (*Figure 5C,D*). The further decondensation of VC heterochromatin after H1 depletion during the late microspore stage suggests the involvement of other factors in the VC. To test whether H1 is sufficient to induce chromatin condensation in microspores, we used the late-microspore-specific *MSP1* promoter (*Honys et al., 2006*) to drive H1 expression (*pMSP1::H1.1-mRFP*, short as *pMSP1::H1*). In *pMSP1::H1*, we observed strong chromocenters in the majority (68%) of late microspores (*Figure 5D*). H1 expression in *pMSP1::H1* is specific to late microspores, and co-localizes with induced chromocenters (*Figure 5E*). These results show that H1 is sufficient to promote heterochromatic foci in late microspores, thus demonstrating the causal relationship between H1 depletion and the decondensation of heterochromatin.

## Discussion

Epigenetic reactivation of TEs in the VC of flowering plants is an intriguing phenomenon, which is important not only for understanding sexual reproduction, but also for elucidating epigenetic silencing mechanisms. Here we show that *Arabidopsis* VC-activated TEs are heterochromatic, and mostly subject to DME-directed demethylation at their TSS (*Figure 1F*). Given the well-demonstrated role of DNA methylation at the TSS for transcriptional suppression (*Barau et al., 2016*; *Eichten et al., 2012*; *Hollister and Gaut, 2009*; *Manakov et al., 2015*; *Meng et al., 2016*), our data demonstrate that DME-mediated demethylation in the VC is the primary cause of TE activation. As DNA demethylation of TEs during reproduction also occurs in rice and maize (*Park et al., 2016*; *Rodrigues et al., 2013*; *Zhang et al., 2014*), species that diverged from *Arabidopsis* more than 150 million years ago (*Chaw et al., 2004*), our results suggest that TE activation in the VC is prevalent among flowering plants.

DME demethylates about ten thousand loci in the VC and central cell, respectively, however, only half of these loci overlap (*Ibarra et al., 2012*). It was unclear why DME targets differ in these cell types, but differences in chromatin configuration have been postulated to contribute (*Feng et al., 2013*). Our finding that the access of DME to heterochromatic TEs in the VC is permitted by the lack of H1 supports this idea. H1 is presumably present in the central cell (*Frost et al., 2018*) but is absent in the VC (*Hsieh et al., 2016*), thus rendering heterochromatic TEs more accessible in the VC. Differential distribution of other factors in the VC and central cell, such as histone variant H3.1

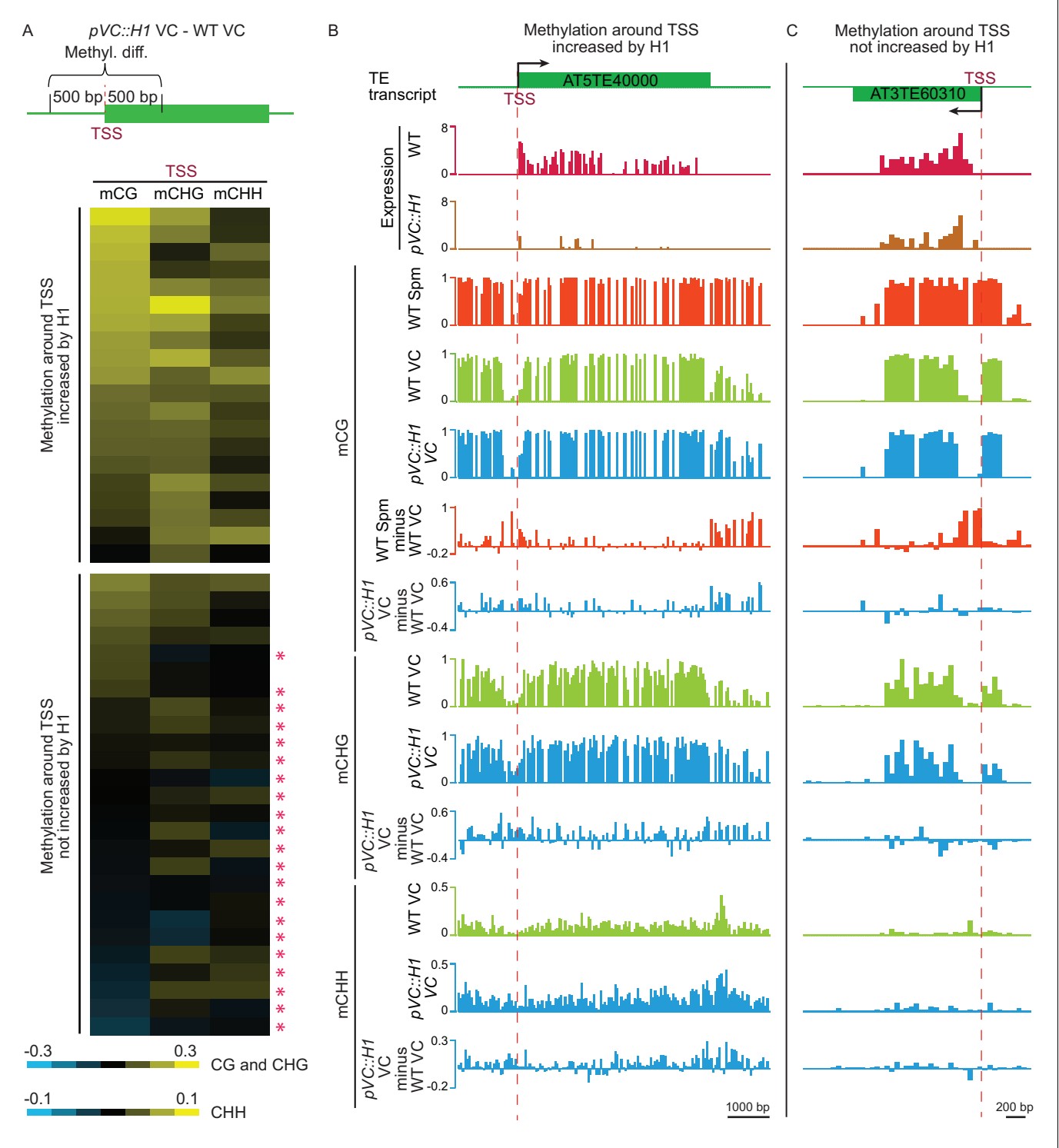

**Figure 4.** H1 suppresses TEs in the vegetative cell via two mechanisms. (A) Heat map demonstrating DNA methylation differences between *pVC::H1* and WT VCs within 500 bp of the TSS of H1-repressed TEs. Asterisks mark TEs whose suppression is not caused by changes in DNA methylation. Data are sorted in descending order based on CG methylation difference for upper and lower panels, respectively. (B,C) Snapshots showing the expression and DNA methylation of representative TEs suppressed by *pVC::H1* via methylation-dependent (B) and -independent (C) mechanisms, respectively. Spm, sperm.

DOI: https://doi.org/10.7554/eLife.42530.015

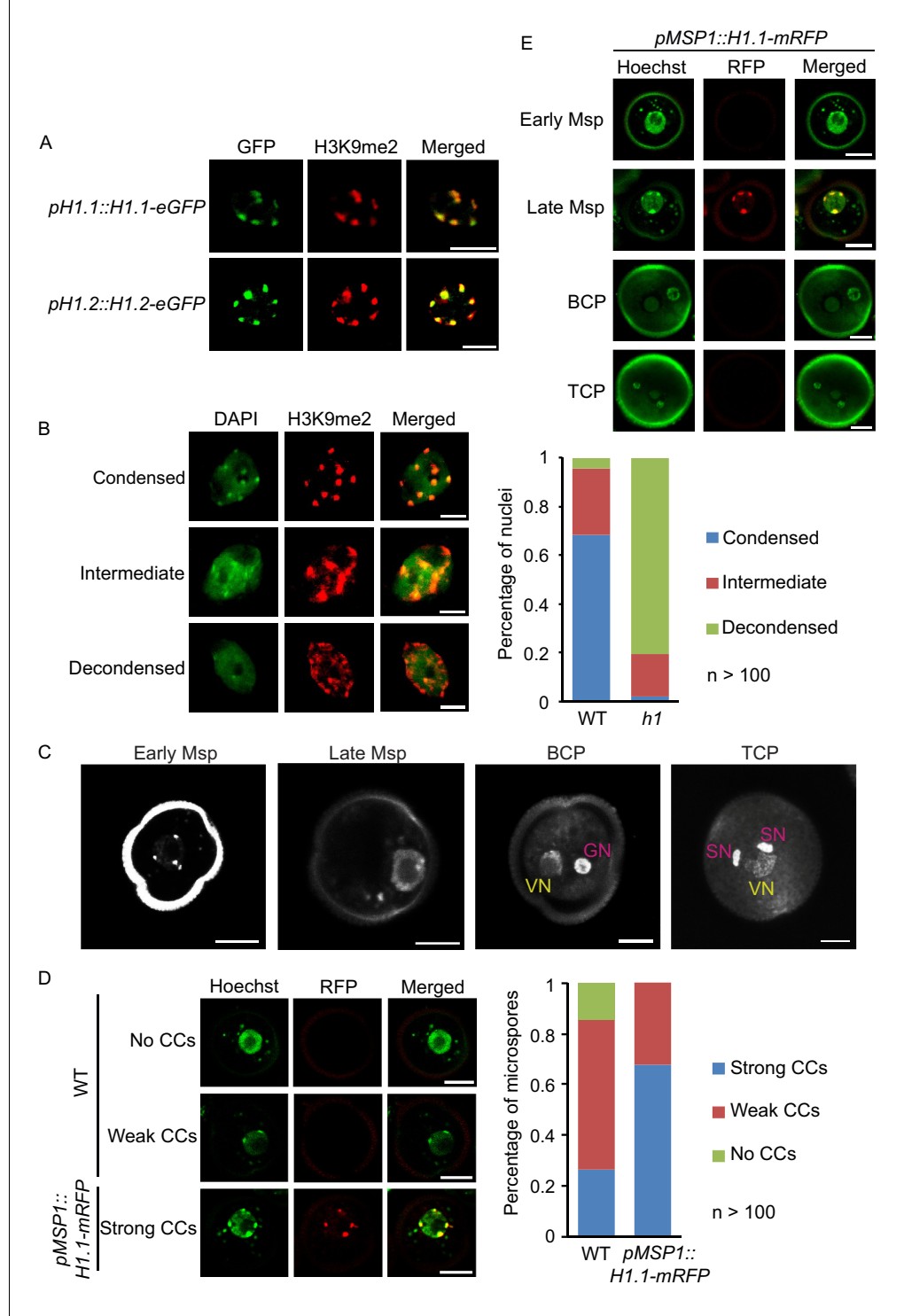

**Figure 5.** Depletion of H1 decondenses heterochromatin in leaves and late microspores. (**A**) Immunostaining with GFP and H3K9me2 antibodies showing the co-localization of H1 and H3K9me2-enriched chromocenters. (**B**) Examples of leaf nuclei with condensed, intermediate or decondensed chromocenters, and their percentages in WT or the *h1* mutant. (**C**) Gradual decondensation of heterochromatin during male gametogenesis in *Arabidopsis*. Micrographs of Hoechst-stained microspores (Msp) and pollen (BCP, bicellular pollen; TCP, tricellular pollen) demonstrate a gradual dispersion of chromocenters in late microspores and subsequently the vegetative nucleus (VN) in pollen. Chromocenters are not detected in the VN of BCP and TCP. GN, generative nucleus. (**D**) Percentages of late microspores with no, weak or strong chromocenters (CCs; examples on the left) in WT and

*Figure 5 continued on next page*

**Figure 5 continued**

*pMSP1::H1.1-mRFP*, in which H1 co-localizes with the strong CCs. (E) H1 is induced only in late microspores in *pMSP::H1.1-mRFP*, and co-localizes with strong CCs. All bars, 5 μm.

DOI: https://doi.org/10.7554/eLife.42530.016

---

(*Borg and Berger, 2015*; *Ingouff et al., 2010*), may also affect DME targeting. Consistently, FACT is required for DME activity in the central cell at many loci even in the absence of H1, whereas DME is entirely independent of FACT in the VC (*Frost et al., 2018*), suggesting the presence of impeding factor(s) other than H1 in the central cell. With distinct chromatin architectures, the vegetative and central cells are excellent systems for understanding how chromatin regulates DNA demethylation.

Our finding that histone H1 affects DME activity adds to the emerging picture of H1 as an important and complex regulator of eukaryotic DNA methylation. H1 depletion causes local hypomethylation in mouse cells (*Fan et al., 2005*) and extensive hypermethylation in the fungi *Ascobolus immersus* (*Barra et al., 2000*) and *Neurospora crassa* (*Seymour et al., 2016*). In *Arabidopsis*, loss of H1 causes global heterochromatic hypermethylation in all sequence contexts by allowing greater access of DNA methyltransferases (*Lyons and Zilberman, 2017*; *Zemach et al., 2013*). Our results suggest that H1 may also influence DME-homologous demethylases that control methylation in somatic tissues (*He et al., 2011*). By regulating both methylation and demethylation, H1 may serve as an integrator of methylation pathways that tunes methylation up or down depending on the locus.

Our data also indicate that the regulatory functions of H1 extend beyond DNA methylation in plants. Activated TEs in the VC can be categorized into four groups, based on the mechanism of their activation (*Figure 6*). TEs in Group I are the least heterochromatic and their activation is dependent on DME but not H1 (*Figures 3H* and *6*). Group II comprises TEs in which H1 absence is required for DME demethylation and activation (*Figure 6*). For TEs in Group III, H1 depletion and DME demethylation are both required for activation, but DME activity is not affected by H1 (*Figure 6*). Group IV TEs are activated by H1 depletion and are not targeted by DME (*Figure 6*). Groups III and IV demonstrate that H1 can silence TEs independently of DNA methylation. Group III also demonstrates that DNA methylation and H1 cooperate to suppress TE expression in plants. Thus, H1 regulates TEs via DNA methylation-dependent and -independent mechanisms.

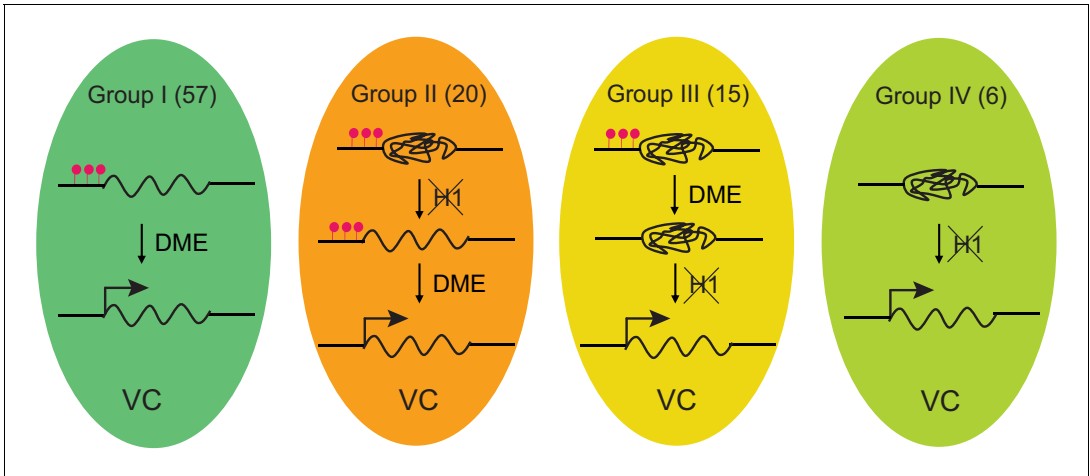

**Figure 6.** Model depicting four mechanisms underlying TE activation in the VC. The number of TEs in each group is shown on the top. Significantly less heterochromatic than TEs in other groups (*Figure 3H*), Group I TEs are activated by DME-directed DNA demethylation. Group II TEs rely on H1 depletion to allow DME demethylation and activation. Group III TEs are demethylated by DME but require H1 depletion to allow transcription (ie. *pVC::H1* represses these TEs without affecting DME). Group IV TEs are not demethylated by DME; their activation is solely dependent on the depletion of H1. TEs belong to each group are listed in *Figure 1—source data 1*. Red lollipops denote DNA methylation.
DOI: https://doi.org/10.7554/eLife.42530.017

During the ongoing arms race between TEs and their hosts, it may be difficult to determine whether TE expression represents temporary TE triumphs or is domesticated by the host to serve a function. TE activation in the VC – a cell that engulfs the male plant gametes – has been proposed as a defense strategy, which generates small RNAs that enhance TE silencing in sperm (*Calarco et al., 2012*; *Ibarra et al., 2012*; *Martínez et al., 2016*; *Slotkin et al., 2009*). However, TEs can also use companion cells as staging grounds for invasion of the gametes (*Wang et al., 2018*). Our demonstration that programmed DME demethylation, which is facilitated by developmental heterochromatin decondensation, is the predominant cause of VC TE activation is consistent with a defensive, host-beneficial model. Nonetheless, the alternative TE-driven model is also plausible. DME demethylates about ten thousand loci in the VC, most of which are small and euchromatic TEs (*Ibarra et al., 2012*). However, only a hundred heterochromatic TEs are activated, at least partially permitted by developmental H1 depletion. As small euchromatic TEs tend to be next to genes, DME demethylation regulates genes and is important for pollen fertility (*Choi et al., 2002*; *Ibarra et al., 2012*; *Schoft et al., 2011*). Our data show that developmental H1 depletion is also important for pollen fertility. Therefore, at least some TEs may be hijacking an essential epigenetic reprogramming process. TE activation in the VC may facilitate both host defense and transposition, with the balance specific to each TE family and changing over evolutionary time. The effects of VC TE activation on TE proliferation in the progeny may warrant investigation, particularly in out-crossing species with aggressive TEs and in natural populations.

# Materials and methods

## Key resources table

| Reagent type (species) or resource | Designation | Source or reference | Identifiers | Additional information |
|---|---|---|---|---|
| Gene (*Arabidopsis thaliana*) | H1.1 | NA | AT1G06760 | |
| Gene (*A. thaliana*) | H1.2 | NA | AT2G30620 | |
| Gene (*A. thaliana*) | DEMETER | NA | AT5G04560 | |
| Genetic reagent (*A. thaliana*) | h1.1–1 | Nottingham Arabidopsis Stock Centre | SALK_128430C | |
| Genetic reagent (*A. thaliana*) | h1.2–1 | GABI-Kat | GABI_406H11 | |
| Genetic reagent (*A. thaliana*) | dme-7 | Nottingham Arabidopsis Stock Centre | SALK_107538 | |
| Genetic reagent (*A. thaliana*) | pLAT52::H1.1-mRFP | this paper | | *LAT52* promoter from tomato is used |
| Genetic reagent (*A. thaliana*) | pMSP1::H1.1-mRFP | this paper | | *MSP1* (AT5G59040) promoter is used |
| Biological sample (*A. thaliana*) | sperm nuclei from Col-0 | this paper | | bisulfite-sequencing |
| Biological sample (*A. thaliana*) | vegetative nuclei from Col-0 | this paper | | bisulfite-sequencing |
| Biological sample (*A. thaliana*) | vegetative nuclei from *pLAT52::H1.1-mRFP* | this paper | | bisulfite-sequencing |
| Biological sample (*A. thaliana*) | pollen from Col-0 | this paper | | RNA-sequencing |
| Biological sample (*A. thaliana*) | pollen from *pLAT52::H1.1-mRFP* | this paper | | RNA-sequencing |
| Biological sample (*A. thaliana*) | pollen from *dme-7/+* | this paper | | RNA-sequencing |

*Continued on next page*

*Continued*

| Reagent type (species) or resource | Designation | Source or reference | Identifiers | Additional information |
|---|---|---|---|---|
| Antibody | GFP | Abcam | Cat# ab290, RRID:AB_303395 | (1:100) |
| Antibody | H3K9me2 | Abcam | Cat# ab1220, RRID:AB_449854 | (1:100) |
| Antibody | Alexa 488- secondaries | ThermoFisher Scientific | Cat# A-11034, RRID:AB_2576217 | (1:200) |
| Antibody | Alexa 555- secondaries | ThermoFisher Scientific | Cat# A28180, RRID:AB_2536164 | (1:200) |
| Software, algorithm | Mikado | *Venturini et al., 2018* | RRID:SCR_016159 | |
| Software, algorithm | Kallisto | *Bray et al., 2016* | RRID:SCR_016582 | |
| Software, algorithm | Sleuth | *Pimentel et al., 2017* | RRID:SCR_016883 | |
| Other | Hoechst 33342 | ThermoFisher Scientific | Cat# H3570 | (1:1000) |
| Other | DAPI | ThermoFisher Scientific | Cat# D1306, RRID:AB_2629482 | (100 ng/mL) |

## Plant materials and growth conditions

*A. thaliana* plants were grown under 16 hr light/8 hr dark in a growth chamber (20°C, 80% humidity). All plants used are of the Col-0 ecotype. *pH1.1::H1.1-eGFP*, *pH1.2::H1.2-eGFP*, *dme-7*, and the *h1* (*h1.1 h1.2* double) mutant lines were described previously (*Schoft et al., 2011*; *She et al., 2013*; *Zemach et al., 2013*). *pLAT52::H1.1-mRFP* and *pMSP1::H1.1-mRFP* were constructed with MultiSite Gateway System into the destination vector pK7m34GW (Invitrogen). The BP clones pDONR-P4-P1R-*pLAT52* and pDONR-P2R-P3-mRFP were kindly provided by Prof. David Twell (Leicester University, UK) (*Eady et al., 1994*). *MSP1* promoter was cloned into pDONR-P4-P1R as described previously (*Honys et al., 2006*). WT plants were transformed via floral dip (*Clough and Bent, 1998*), and T2 or T3 plants homozygous for the transgene were used in this study.

## Pollen extraction, RNA sequencing and quantitative RT-PCR

Open flowers were collected for pollen isolation in Galbraith buffer (45 mM $MgCl_2$, 30 mM sodium citrate, 20 mM MOPS, 1% Triton-X-100, pH 7.0) by vortexing at 2000 rpm for 3 min. The crude fraction was filtered through a 40 μm cell strainer to remove flower parts, and subsequently centrifuged at 2600 g for 5 min to obtain pollen grains. RNA was extracted from pollen grains with RNeasy Micro Kit (Qiagen) following manufacturer's instructions. RNA-sequencing libraries were prepared using Ovation RNA-seq Systems 1–16 for Model Organisms (Nugen Technologies), and sequenced on the Hiseq 2500 (Illumina) instrument at the UC Berkeley Vincent J. Coates Genomics Sequencing Laboratory or on the Nextseq 500 (Illumina) at the John Innes Centre. Quantitative RT-PCR (qRT-PCR) was performed as described previously (*Walker et al., 2018*), and *TUA2* was used as an internal control. Primers for qRT-PCR are listed in *Supplementary file 2*.

## In vitro pollen germination

The experiment was performed as described previously with some modifications (*Rodriguez-Enriquez et al., 2013*). Pollen from three newly opened flowers for individual plants were brushed on cellulose membrane sitting on germination medium (18% sucrose, 0.01% boric acid, 1 mM CaCl2, 1 mM Ca(NO3)2, 1 mM KCl, 0.25 mM spermidine, pH 8.0 with KOH adjusted, 0.5% agarose) in small petri dishes. Petri dishes were placed in a box with a piece of wet tissue at the bottom to keep humidity. The boxes were incubated in Sanyo cabinet (MLR-351H) at 21°C for 4 hr. 300 pollen grains were counted from each cellulose membrane and eight replicates for each genotype. Pollen germination was counted by using ImageJ.

## RNA-seq analysis

TE transcript annotation was created using RNA-seq data from four biological replicates of pollen. Tophat2, Hisat, and STAR were used to align RNA-seq reads to the TAIR10 genome, and transcripts were assembled using CLASS2, StringTie, and Cufflinks, respectively. Assembled transcripts were selected by Mikado using default options except that the BLAST and Transdecoder steps were disabled (*Venturini et al., 2018*). As a result, 21381 transcripts (called superloci; GSE120519) were identified.

To identify VC-activated TEs, we first refined the list of superloci by selecting those overlapping with TAIR10 TE annotation. Subsequently to eliminate TE-like genes from the refined list, superloci with CG methylation less than 0.7 in rosette leaves (*Stroud et al., 2014*; *Stroud et al., 2013*) were excluded. This gave rise to an annotation of pollen TE transcripts, which was combined with TAIR10 gene annotation for Kallisto analysis (*Bray et al., 2016*). RNA-seq data from WT and *dme*/+ pollen (this study) and rosette leaves (*Walker et al., 2018*), each including three biological replicates, were processed using Kallisto and Sleuth (*Bray et al., 2016*; *Pimentel et al., 2017*). TEs that are transcribed at least five times more in WT pollen than leaves (with p<0.05, likelihood ratio test) are considered as activated in the VC (refer to *Figure 1—source data 1* for the list of VC-activated TEs). A total of 2845 genes were found to be expressed in pollen with TPM (transcripts per million) more than five in the Kallisto output (*Figure 3—source data 1*; data used in *Figure 3B*).

To identify TEs and genes that are suppressed by H1 in the VC, we analyzed RNA-seq data from WT and *pLAT52::H1.1-mRFP line #2* (short as *pVC::H1* unless specified otherwise) pollen using Kallisto and Sleuth as described above. Significant differential expression was defined with a fold change at least two and a p-value less than 0.05. H1-repressed TEs were listed in *Figure 1—source data 1*.

## Whole-genome bisulfite sequencing and analysis

Vegetative and sperm nuclei were isolated via FACS as described previously (*Ibarra et al., 2012*). Bisulfite-sequencing libraries were prepared as previously described (*Walker et al., 2018*). Sequencing was performed on Hiseq 2500 (Illumina) at the UC Berkeley Vincent J. Coates Genomics Sequencing Laboratory, Hiseq 4000 (Illumina) at Novogene Ltd. and Harvard University, and Nextseq 500 (Illumina) at Cambridge University Biochemistry Department and the John Innes Centre. Sequenced reads (100, 75, or 50 nt single-end) were mapped to the TAIR10 reference genome, and cytosine methylation analysis was performed as previously described (*Ibarra et al., 2012*).

## Identification of DME targets and H1 hyperDMRs in the VC

As all CG hypomethylation in the VC in comparison to sperm is caused by DME (*Ibarra et al., 2012*), we identified VC DME targets via detecting CG differentially methylated regions (DMRs) that are hypermethylated in sperm in comparison to the VC. DMRs were identified first by using MethPipe (settings: p=0.05 and bin = 100) (*Song et al., 2013*), and subsequently retained if the fractional CG methylation across the whole DMR was at least 0.2 higher in sperm than the VC. The refined DMRs were merged to generate larger DMRs if they occurred within 300 bp. Finally, merged DMRs were retained if they cover at least 100 bp, and the fractional CG methylation across the whole DMR was significantly (Fisher's exact test p<0.01) and substantially (>0.2) higher in sperm than the VC. This resulted in the identification of 11896 VC DME targets (*Figure 1—source data 2*).

H1 hyperDMRs were identified using the same criteria, except comparing CG methylation in VCs from *pVC::H1* and WT. In total, 2964 H1 hyperDMRs were identified (*Figure 2—source data 1*).

## Box plots

Box plots compare the enrichment of genomic or chromatin features among TEs (*Figures 1B* and *3G*, *Figure 1—figure supplement 1B*) or VC DME targets (*Figure 2I*) as described in corresponding figure legends. ChIP-seq data for H3K9me2 (*Stroud et al., 2014*), and ChIP-chip data for H1 (*Rutowicz et al., 2015*), H3K27me3 (*Kim et al., 2012*), and other histone modifications (*Roudier et al., 2011*) were used.

## Density plots

All DNA methylation kernel density plots compare fractional methylation within 50 bp windows. We used windows with at least 20 informative sequenced cytosines and fractional methylation of at least 0.5 (*Figure 2D*, *Figure 2—figure supplement 2*) or 0.7 (*Figure 2E*) for CG context, and 0.4 and 0.1 for CHG and CHH context, respectively, in at least one of the samples being compared.

## Meta analysis (ends analysis)

Ends analysis for TEs and genes was performed as described previously (*Ibarra et al., 2012*). Similarly, ends analysis of TE transcripts was performed using the annotation of VC-activated TEs described above (*Figure 1—source data 1*). DNA methylation data from *Ibarra et al. (2012)* was used.

In *Figure 2J*, DME sites were aligned at the most demethylated cytosine, and average CG methylation levels for each 10 bp interval at both sides were plotted. To identify individual hypomethylation sites created by DME, we first obtained the 50 bp windows with a CG methylation difference larger than 0.5 between sperm and VC (sperm – VC >0.5 and Fisher's exact test p<0.001). Windows were then merged if they occurred within 200 bp. Merged windows were retained for further analysis if the fractional CG methylation across the whole site was 0.2 greater in sperm than VC (sperm – VC >0.2 and Fisher's exact test p<0.0001). This resulted in 13610 DME sites, which were separated into five groups according to H3K9me2 level (*Stroud et al., 2014*):<2.5, 2.5–4.3, 4.3–6.5, 6.5–10.5, and >10.5 (*Figure 2J*). The most demethylated cytosine within each site was identified if it had the greatest differential methylation in sperm than VC among cytosines in the CG context (sperm – VC >0.2, and Fisher's exact test p<0.001) and was sequenced at least 10 times.

## DNA methylation analysis of H1-repressed TEs

Differential methylation at a 1000 bp region centered upon the TSS of H1-repressed TEs was calculated between VCs of *pVC::H1* and WT (*Figure 4A*). TEs whose differential methylation is significant (Fisher's exact test p<0.001) and larger than 0.2 (in CG context), 0.1 (in CHG context), or 0.05 (in CHH context) are illustrated in the upper panel in *Figure 4A*.

## Confocal and scanning electron microscopy

Microspores and pollen were isolated as described previously (*Borges et al., 2012*), stained with Hoechst or DAPI, and examined under a Leica SP8 confocal microscope. Scanning electron microscopy was performed on a Zeiss Supra 55 VP FEG.

## Immunofluorescence

Immunofluorescence was performed as described previously with small modifications (*Yelagandula et al., 2014*). Rosette leaves from 3-week-old plants were fixed in TRIS buffer with 4% paraformaldehyde (10 mM Tris-HCl pH 7.5, 10 mM EDTA, 100 mM NaCl) for 20 min. After being washed with TRIS buffer twice, the fixed leaves were chopped with razor blades in 1 mL of lysis buffer (15 mM Tris pH 7.5, 2 mM EDTA, 0.5 mM spermine, 80 mM KCl, 20 mM NaCl, 0.1% Triton X-100) and filtered through a 35 μm cell strainer. Nuclei were pelleted via centrifugation at 500 g for 3 min and resuspended in 100 μL of lysis buffer. Next, 10 μL was spotted onto coverslips, air-dried, and post-fixed in PBS with 4% paraformaldehyde for 30 min. After being washed with PBS twice, coverslips were incubated in blocking buffer (PBS with 1% BSA) at 37°C for 30 min and then incubated in blocking buffer with primary antibodies at 4°C overnight (Mouse anti-H3K9me2 Abcam ab1220, 1:100; Rabbit anti-GFP Abcam ab290, 1:100). After being washed with PBS three times, coverslips were incubated in PBS with secondary antibodies at 37°C for 30 min, and then washed with PBS three times again before being counterstained and mounted in Vectashield mounting media with DAPI (Vector H-1200).

## Acknowledgements

We thank David Twell for the pDONR-P4-P1R-pLAT52 and pDONR-P2R-P3-mRFP vectors, the John Innes Centre Bioimaging Facility (Elaine Barclay and Grant Calder) for their assistance with microscopy, and the Norwich BioScience Institute Partnership Computing infrastructure for Science Group

for High Performance Computing resources. This work was funded by a Biotechnology and Biological Sciences Research Council (BBSRC) David Phillips Fellowship (BB/L025043/1; SH, JZ and XF), a European Research Council Starting Grant ('SexMeth' 804981; XF) and a Grant to Exceptional Researchers by the Gatsby Charitable Foundation (SH and XF).

## Additional information

### Funding

| Funder | Grant reference number | Author |
|---|---|---|
| Biotechnology and Biological Sciences Research Council | BBL0250431 | Jingyi Zhang Xiaoqi Feng |
| Gatsby Charitable Foundation | | Shengbo He Xiaoqi Feng |
| H2020 European Research Council | 804981 SexMeth | Xiaoqi Feng |

The funders had no role in study design, data collection and interpretation, or the decision to submit the work for publication.

### Author contributions

Shengbo He, Conceptualization, Data curation, Formal analysis, Validation, Investigation, Visualization, Methodology, Writing—original draft; Martin Vickers, Data curation, Data analysis; Jingyi Zhang, Methodology, Acquisition of data; Xiaoqi Feng, Conceptualization, Resources, Supervision, Funding acquisition, Investigation, Methodology, Project administration, Writing—review and editing

### Author ORCIDs

Shengbo He (iD) https://orcid.org/0000-0003-3773-9995
Xiaoqi Feng (iD) https://orcid.org/0000-0002-4008-1234

### Decision letter and Author response

Decision letter https://doi.org/10.7554/eLife.42530.034
Author response https://doi.org/10.7554/eLife.42530.035

## Additional files

### Supplementary files

• Supplementary file 1. Sequencing summary statistics for bisulfite sequencing libraries. Mean DNA methylation (Met) was calculated by averaging methylation of individual cytosines in each context, and chloroplast CHH methylation was used as a measure of cytosine non-conversion and other errors. SN, sperm nuclei; VN, vegetative nuclei.
DOI: https://doi.org/10.7554/eLife.42530.018

• Supplementary file 2. List of primers for quantitative RT-PCR.
DOI: https://doi.org/10.7554/eLife.42530.019

• Transparent reporting form
DOI: https://doi.org/10.7554/eLife.42530.020

### Data availability

We have deposited our sequencing data in GEO (GSE120519). Source data files have been provided for Figures 1 to 3.

The following dataset was generated:

| Author(s) | Year | Dataset title | Dataset URL | Database and Identifier |
|---|---|---|---|---|
| He S, Vickers M, Zhang J, Feng X | 2018 | Developmental depletion of histone H1 in Arabidopsis sex cells causes DNA demethylation, heterochromatin decondensation and transposon activation | https://www.ncbi.nlm.nih.gov/geo/query/acc.cgi?acc=GSE120519 | NCBI Gene Expression Omnibus, GSE120519 |

The following previously published datasets were used:

| Author(s) | Year | Dataset title | Dataset URL | Database and Identifier |
|---|---|---|---|---|
| Nishimura T, Zilberman D | 2012 | Active DNA demethylation in plant companion cells reinforces transposon methylation in gametes | http://www.ncbi.nlm.nih.gov/geo/query/acc.cgi?acc=GSE38935 | NCBI Gene Expression Omnibus, GSE38935 |
| Wilczynski B, Rutowicz K | 2015 | Analysis of genomic distribution of three linker histone variants in Arabidopsis under normal and low light conditions, by ChIP-chip | https://www.ebi.ac.uk/arrayexpress/experiments/E-MTAB-2804/ | ArrayExpress, E-MTAB-2804 |
| Stroud H | 2013 | Non-CG methylation patterns shape the epigenetic landscape in Arabidopsis | http://www.ncbi.nlm.nih.gov/geo/query/acc.cgi?acc=GSE51304 | NCBI Gene Expression Omnibus, GSE51304 |
| Zilberman D, Nishimura T | 2012 | EMF1 and PRC2 Cooperate to Repress Key Regulators of Arabidopsis Development | http://www.ncbi.nlm.nih.gov/geo/query/acc.cgi?acc=GSE34689 | NCBI Gene Expression Omnibus, GSE34689 |
| Colot V, Roudier F, Martin-Magniette M | 2011 | Establishing a reference epigenome in arabidopsis seedlings | http://www.ncbi.nlm.nih.gov/geo/query/acc.cgi?acc=GSE24710 | NCBI Gene Expression Omnibus, GSE24710 |

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
