## [Decision Letter]

Thank you for submitting your article "Natural depletion of H1 in sex cells causes DNA demethylation, heterochromatin decondensation and transposon activation" for consideration by *eLife*. Your article has been reviewed by two peer reviewers, one of whom is a member of our Board of Reviewing Editors, and the evaluation has been overseen by Christian Hardtke as the Senior Editor.

The reviewers have discussed the reviews with one another and the Reviewing Editor has drafted this decision to help you prepare a revised submission.

For a long time, it has been known that there is a reprogramming of epigenetic marks in both the vegetative nucleus and the sperm nuclei. For example, it was known that in the vegetative nuclei there is a loss of H3K9me2, lack of activity of the chromatin remodeler DDM1 and high activity of the DNA demethylase DME and of the de novo methyltransferase DRM2. This manuscript adds new knowledge about the intricacy of the mechanisms leading to TE reactivation in the vegetative nucleus. The manuscript is well written, concise, and sincere in the presentation of the results including pointing to the limits of the analysis. If the issues raised below can be properly addressed, the manuscript should be suitable for publication in *eLife*.

Essential revisions:

1) TE reactivation observed by the authors is very modest compared to the analysis of DNA methylation for the VC that has been published before (1781 TEs in Calarco et al., 2012, Cell vs. 114 TEs on this manuscript). I think that this difference must be discussed. At the same time, in previous manuscripts it has been shown that companion cell demethylation mediated by DME takes place at loci that tend to be within smaller AT-rich TEs that are enriched for euchromatin-associated histone modifications. This is quite the opposite of what the authors find on the reactivated TEs and should be discussed in order to connect better with DME function.

2) Another point regarding the 114 reactivated TEs. In terms of proximity to genes these TEs are located in pericentromeric regions where genes and TEs coexist. Is this a feature of these TEs? Could they be involved in gene regulation? Do they give rise to more or a different profile of small RNAs compared to other TEs? Are these sRNAs more mobile to the SCs? This analysis could easily be added since datasets are already available and will help to dig into the characteristics of theses TEs and why they are targets of DME.

3) Because a very important function of DME is the regulation of imprinted genes I think it would be interesting to the overall community for the authors to also include an analysis (or at least comment) on the effects of H1 overexpression on imprinted genes. In principle DME and related glycosylases preferentially demethylate gene-adjacent sequences so this might be an important part of how it regulates gene expression.

4) It is intriguing why the authors use two different transgenic lines for H1 overexpression: one with the *LAT52* promoter and a different one with the *MSP1* promoter. Since the LAT52 promoter is active at later stages of pollen development, are chromocenters also seen at later stages of pollen development in the *pLAT52::H1.1-RFP* line? Can this be quantified and added to the manuscript along with a more detail about why the two different promoters were used?

Two relatively major issues in writing:

1)"Examination of DNA methylation in VCs from *dme*/+ heterozygous 155 plants (*dme* homozygous mutants are embryonic lethal), which produce a 50:50 ratio of *dme* 156 mutant and WT pollen, REVEALED PARTIAL RESTORATION of methylation at TSS and TTS of VC – 157 activated TEs (Figure 1D,E)."

Do you really mean "restoration"? Restoration would mean there was demethylation in the VC and then DNA methylation was added back. If in the *dme* VCs, there was simply not as much loss of DNA methylation, do not use the word restoration.

2) The following sentence (subsection “Depletion of H1 decondenses heterochromatin during male gametogenesis”) does not "make sense."

"We observed strong and weak chromocenters in 27% and 59%, respectively, of late microspore nuclei, whereas no chromocenters were observed in the VC at either bicellular or tricellular pollen stage (Figure 5C, Figure 5—figure supplement 1).”

"respectively" is used to match A and B with C and D. In the sentence there is no D (see below).

“we observed strong and weak chromocenters in 27% (A) and 59% (B), respectively, of late microspore nuclei (C).”

Something is missing in this sentence.

---

## [Author Response]

*For a long time, it has been known that there is a reprogramming of epigenetic marks in both the vegetative nucleus and the sperm nuclei. For example, it was known that in the vegetative nuclei there is a loss of H3K9me2, lack of activity of the chromatin remodeler DDM1 and high activity of the DNA demethylase DME and of the* de novo *methyltransferase DRM2. This manuscript adds new knowledge about the intricacy of the mechanisms leading to TE reactivation in the vegetative nucleus. The manuscript is well written, concise, and sincere in the presentation of the results including pointing to the limits of the analysis. If the issues raised below can be properly addressed, the manuscript should be suitable for publication in eLife.*

We are delighted for the positive valuation from the reviewers and the reviewing editor, and grateful for their constructive comments and suggestions. In accordance to these valuable comments, we have performed additional experiments and analyses, and revised the manuscript.

Beyond the comments we received, we have further validated the causal relationship between DME demethylation and TE activation in the VC through performing pollen RNA-seq from *dme*/+ heterozygous plants (*dme* homozygous mutant is embryonic lethal). This additional data has been added as Figure 1G, and to the Figure 1—source data 1 and the text: “Consistently, RNA-seq of pollen from *dme*/+ heterozygous plants showed significantly reduced expression at the 114 VC-activated TEs (Figure 1G, Figure 1—source data 1). As *dme*/+ heterozygous plants produce half *dme* mutant and half WT pollen, we expect the transcription of VC-activated TEs to be reduced to roughly half in *dme*/+ pollen. The 114 VC-activated TEs are transcribed at levels close to this expectation (Figure 1G), with the median ratio of their transcription in *dme*/+ versus WT pollen being 0.43 (Figure 1—source data 1).”

We believe this result strengthens one of the major conclusions of our manuscript, i.e. TE activation in the VC is caused by DME-directed DNA demethylation.

Essential revisions:1) TE reactivation observed by the authors is very modest compared to the analysis of DNA methylation for the VC that has been published before (1781 TEs in Calarco et al., 2012, Cell vs. 114 TEs on this manuscript). I think that this difference must be discussed. At the same time, in previous manuscripts it has been shown that companion cell demethylation mediated by DME takes place at loci that tend to be within smaller AT-rich TEs that are enriched for euchromatin-associated histone modifications. This is quite the opposite of what the authors find on the reactivated TEs and should be discussed in order to connect better with DME function.

We thank the reviewers and editor for the helpful comment. The 1781 TEs were identified as overlapping with 2270 VC-sperm CHH DMRs, almost all of which are hypermethylated in the VC in comparison to sperm (Calarco et al., 2012). Therefore, these TEs/DMRs should not be related to DME-directed demethylation or TE activation in the VC. Indeed, only 6 of the 2270 CHH DMRs overlap DME targets, and only 6 out of the 1781 TEs are activated in pollen.

However, there is indeed a contrast between the large number of small euchromatic TEs demethylated by DME in the VC and the small number of heterochromatic TEs we found actually activated. We thank the reviewers for pointing this out and agree that this is indeed worth more discussion, because one of the significant contributions of our manuscript is revealing the actual modest scale of TE activation in the VC. However, first of all, it is worth to note that DME demethylates both small euchromatic TEs and the edges of long heterochromatic TEs (Supplementary Figure 9G and H in Ibarra et al. 2012; Frost et al. 2018), although the latter are less in their number. For transcriptional regulation, the methylation at TE edges is most important as they tend to overlap the TSS.

Our results demonstrate that only 114 heterochromatic TEs in pericentromeric regions are activated in the VC, as a result of DME demethylation and developmental H1 depletion. This supports the “TE-driven model” we proposed in the last paragraph of the Discussion, which we have now expanded: “DME demethylates about ten thousand loci in the VC, most of which are small and euchromatic TEs (Ibarra et al., 2012). However, only a hundred heterochromatic TEs are activated, at least partially permitted by developmental H1 depletion. As small euchromatic TEs tend to be next to genes, DME demethylation regulates genes and is important for pollen fertility (Choi et al., 2002; Ibarra et al., 2012; Schoft et al., 2011). Our data show that developmental H1 depletion is also important for pollen fertility. Therefore, at least some TEs may be hijacking an essential epigenetic reprogramming process […]”

2) Another point regarding the 114 reactivated TEs. In terms of proximity to genes these TEs are located in pericentromeric regions where genes and TEs coexist. Is this a feature of these TEs?

Yes. As shown in Figure 1A and B and Figure 1–figure supplement 1A and B (subsection “Heterochromatic transposons are preferentially expressed in the vegetative cell”), being heterochromatic and pericentromeric is a significant feature of these reactivated TEs.

Could they be involved in gene regulation?

To understand whether nearby genes are regulated/affected by the activation of these TEs, we identified 70 genes within 1 kb of the 114 reactivated TEs. Analysis of our newly obtained *dme/+* pollen RNA-seq data shows that the transcription of none of the 70 genes is significantly affected in *dme/+* pollen, demonstrating that TE activation in the VC is unlikely involved in gene regulation. This is consistent with previous findings that DME regulates gene expression through the demethylation of small euchromatic TEs that are adjacent to genes, and is now elaborated in the Discussion section as quoted in response to Comment 1, specifically: “As small euchromatic TEs tend to be next to genes, DME demethylation regulates genes and is important for pollen fertility (Choi et al., 2002; Ibarra et al., 2012; Schoft et al., 2011)”.

Do they give rise to more or a different profile of small RNAs compared to other TEs? Are these sRNAs more mobile to the SCs? This analysis could easily be added since datasets are already available and will help to dig into the characteristics of theses TEs and why they are targets of DME.

We are grateful for the thoughtful suggestion. Using published pollen and sperm sRNA-seq data (Slotkin et al., 2009; GSE61028), we find that the 114 reactivated TEs are associated with more sRNAs of 21-24 nt lengths than other TEs in both pollen and sperm (see box plots Author response image 1; all P < 0.01, Kolmogorov–Smirnov test). This result is consistent with the proposal that TE transcription in the VC gives rise to sRNAs that can move into sperm (Slotkin et al., 2009). These 114 activated TEs are associated with more 24 nt sRNAs in sperm than in pollen (see the pie charts Author response image 1), suggesting 24 nt sRNAs might be more mobile. This is different from the previous proposal that the mobile sRNAs are 21 nt sRNAs produced from transcribed TEs (Slotkin et al., 2009), and an alternative explanation also cannot be excluded, i.e. sperm may produce more 24 nt sRNAs from these TEs than the VC. Furthermore, because the differences in the abundance of sRNAs between the reactivated and other TEs are small, and the published data involve no replicates, we do not feel these analyses are conclusive. Neither this manuscript aims to elucidate on sRNA movement between VC and sperm, which we feel require elaborate studies. Therefore, we did not add these analyses to the manuscript.

3) Because a very important function of DME is the regulation of imprinted genes I think it would be interesting to the overall community for the authors to also include an analysis (or at least comment) on the effects of H1 overexpression on imprinted genes. In principle DME and related glycosylases preferentially demethylate gene-adjacent sequences so this might be an important part of how it regulates gene expression.

We appreciate the excellent suggestion, and have performed the requested analysis and revised the manuscript by adding the following to the Results:

“Given the importance of H1 removal for DME-directed DNA demethylation and the well demonstrated role of DME demethylation in regulating gene expression (Choi et al., 2002; Ibarra et al., 2012; Schoft et al., 2011), we investigated the contribution of H1 to gene regulation in pollen. RNA-seq was performed using pollen from the *pVC::H1* line (#2), which showed strong H1 expression in VC (Figures 2B and 3A). […] Consistently, DME targets likely regulating imprinted genes are associated with significantly less H1 in somatic tissues than the VC DME targets that are dependent on H1 (Figure 3C).” (Subsection “H1 represses transposons via methylation-dependent and independent mechanisms”).

4) It is intriguing why the authors use two different transgenic lines for H1 overexpression: one with the LAT52 promoter and a different one with the MSP1 promoter. Since the LAT52 promoter is active at later stages of pollen development, are chromocenters also seen at later stages of pollen development in the pLAT52::H1.1-RFP line? Can this be quantified and added to the manuscript along with a more detail about why the two different promoters were used?

These are excellent questions. As we explained in the manuscript, “To understand how H1 affects DME activity, we ectopically expressed H1 in the VC. To ensure H1 incorporation into VC chromatin, we used the *pLAT52* promoter, which is expressed from the late microspore stage immediately prior to Pollen Mitosis 1, and is progressively upregulated in VC during later stages of pollen development (Eady, Lindsey and Twell, 1994; Grant-Downton et al., 2013). Using *pLAT52* to drive the expression of H1.1 tagged with mRFP (simplified as *pVC::H1*), we observed continuous H1-mRFP signal in the VC at the bicellular and tricellular pollen stages, while the signal was undetectable in the generative cell and sperm (Figure 2B). H1-mRFP signal was also undetectable in late microspores (Figure 2B), probably due to the low activity of *pLAT52* at this stage (Eady et al., 1994).” (Subsection “Vegetative-cell-expressed H1 impedes DME from accessing heterochromatic transposons”)

“*pVC[LAT52]::H1* VC also showed no obvious chromocenters (n > 500; Figure 2B). This suggests either that H1 expression is not strong enough in *pVC[LAT52]::H1*, or other factors are involved in heterochromatin decondensation in the VC. […] These results show that H1 is sufficient to promote heterochromatic foci in late microspores, thus demonstrating the causal relationship between H1 depletion and the decondensation of heterochromatin.” (Subsection “Depletion of H1 decondenses heterochromatin during male gametogenesis”)

The quantification of chromocenters in the *pLAT52::H1.1-RFP* line has been added to the text (bolded and underlined in the quotation).

Two relatively major issues in writing:1)"Examination of DNA methylation in VCs from dme/+ heterozygous 155 plants (dme homozygous mutants are embryonic lethal), which produce a 50:50 ratio of dme 156 mutant and WT pollen, REVEALED PARTIAL RESTORATION of methylation at TSS and TTS of VC – 157 activated TEs (Figure 1D,E)."Do you really mean "restoration"? Restoration would mean there was demethylation in the VC and then DNA methylation was added back. If in the dme VCs, there was simply not as much loss of DNA methylation, do not use the word restoration.

In accordance, the text has been changed to “revealed *an intermediate level* of methylation at TSS and TTS of VC-activated TEs”.

2) The following sentence (subsection “Depletion of H1 decondenses heterochromatin during male gametogenesis”) does not "make sense.""We observed strong and weak chromocenters in 27% and 59%, respectively, of late microspore nuclei, whereas no chromocenters were observed in the VC at either bicellular or tricellular pollen stage (Figure 5C, Figure 5—figure supplement 1).”"respectively" is used to match A and B with C and D. In the sentence there is no D (see below).“we observed strong and weak chromocenters in 27% (A) and 59% (B), respectively, of late microspore nuclei (C).”Something is missing in this sentence.

We apologize for the confusion caused by the ambiguous expression, and have revised the text into “we observed strong and weak chromocenters, respectively, in 27% and 59% of late microspore nuclei […]”.